# Qualitative and Quantitative Resistance against Early Blight Introgressed in Potato

**DOI:** 10.3390/biology10090892

**Published:** 2021-09-10

**Authors:** Pieter J. Wolters, Doret Wouters, Emil J. Kromhout, Dirk Jan Huigen, Richard G. F. Visser, Vivianne G. A. A. Vleeshouwers

**Affiliations:** Plant Breeding, Wageningen University & Research, 6708 PB Wageningen, The Netherlands; jaap.wolters@wur.nl (P.J.W.); doret.wouters@wur.nl (D.W.); emil.kromhout@wur.nl (E.J.K.); dirkjan.huigen@wur.nl (D.J.H.); richard.visser@wur.nl (R.G.F.V.)

**Keywords:** *Solanum*, *S. commersonii* subsp. *malmeanum*, *S. berthaultii*, *S. tuberosum*, *Alternaria solani*, endosperm balance number (EBN), interspecific hybrids, hybrid vigour, necrotroph

## Abstract

**Simple Summary:**

Early blight is a disease of potato caused by the *Alternaria* fungus (notably *A. solani*). Fungicides that are commonly used to protect potato against the disease are losing their effectiveness and an alternative control method is desired. In this research, we identified several relatives of potato from Central and South America that have a high natural resistance against early blight. Although these plants belong to other species, it was possible to cross them with cultivated potato. The resistance was inherited in offspring plants, but, interestingly, the different species seem to contain distinct types of resistance. More detailed studies will help increase our knowledge of the mechanism(s) that cause resistance. Highly resistant offspring plants can be used to develop new potato varieties with a natural resistance to early blight.

**Abstract:**

Early blight is a disease of potato that is caused by *Alternaria* species, notably *A. solani*. The disease is usually controlled with fungicides. However, *A. solani* is developing resistance against fungicides, and potato cultivars with genetic resistance to early blight are currently not available. Here, we identify two wild potato species, which are both crossable with cultivated potato (*Solanum tuberosum*)*,* that show promising resistance against early blight disease. The cross between resistant *S. berthaultii* and a susceptible diploid *S. tuberosum* gave rise to a population in which resistance was inherited quantitatively. *S. commersonii* subsp. *malmeanum* was also crossed with diploid *S. tuberosum*, despite a differing endosperm balance number. This cross resulted in triploid progeny in which resistance was inherited dominantly. This is somewhat surprising, as resistance against necrotrophic plant pathogens is usually a quantitative trait or inherited recessively according to the inverse-gene-for-gene model. Hybrids with high levels of resistance to early blight are present among progeny from *S. berthaultii* as well as *S. commersonii* subsp. *malmeanum*, which is an important step towards the development of a cultivar with natural resistance to early blight.

## 1. Introduction

Early blight, caused by the necrotrophic fungus *Alternaria solani* and closely related *Alternaria* species, is an important disease of potato that can cause major yield losses [1,2,3]. Development of early blight is dependent on plant age and environmental conditions such as higher temperatures, humidity and abiotic stress factors [4,5,6,7]. No resistant potato cultivars are available at the moment and the disease is mainly controlled by the application of fungicides [8]. This is not a sustainable solution, especially since fungicide-resistant strains of *A. solani* are emerging [9,10]. The development of fungicide resistance in *Alternaria* isolates and increased attention on sustainability (limiting the use of fungicides and fertilizer), combined with a changing climate that is favourable for early blight development, means that there is a need for potato cultivars with innate resistance to the disease.

Potato was domesticated in the Andes region in South America and wild potato relatives from the *Solanum* sect. *Petota* are a rich source of genetic and phenotypic diversity [11,12,13]. They have been extensively studied to identify resistance genes against potato pathogens, especially *Phytophthora infestans* [14,15]. Wild potato relatives are a promising source of resistance against *A. solani* too [16,17,18,19,20], but the genes that underly resistance remain to be identified.

Resistance from wild *Solanum* species can be introgressed into cultivated potato through crossbreeding. Modern potato cultivars are usually tetraploids, but wild *Solanum* species can range from diploid to hexaploid. According to the endosperm balance number (EBN) hypothesis, genotypes with different ploidy levels can cross with each other as long as they belong to the same EBN group (Johnston 1980 and 1982). The EBN of many *Solanum* species has been systematically determined by performing crosses with species for which the EBN number is known [21]. In North America, diploids are usually 1 EBN and tetraploids are mostly 2 EBN. In contrast, South American diploids and tetraploids are usually 2 EBN and 4 EBN, respectively [21]. Chromosome doubling or the formation of 2n gametes can be used to make a lower EBN species compatible with a higher EBN species [21]. The production of 2n gametes is not uncommon in *Solanum* [22,23,24].

The tetraploid nature of most cultivated potato complicates the development of new potato cultivars. It is difficult to get rid of unfavourable alleles from wild species when these are used in breeding. Hybrid breeding, using diploid inbred lines, was recently proposed as a promising solution to speed up the development of new potato cultivars [25,26]. In order to make good use of the diversity of wild *Solanum*, it is instrumental to identify the genes or molecular markers linked to genes that underly traits of interest. Resistance to early blight in potato is usually described as being quantitative [6,27,28,29], but detailed studies on the genetics in wild species are lacking.

In this study, we selected two wild *Solanum* genotypes with high levels of resistance to *A. solani*. They were both crossed with susceptible diploid *S. tuberosum* genotypes. Hybrids with high levels of resistance to *A. solani* were identified among progeny of both populations, which is an important step towards using resistant wild *Solanum* germplasm to introgress early blight resistance in cultivated potato. Moreover, we observed two distinct inheritance patterns of resistance to *A. solani*. These findings will help the development of potato cultivars that are naturally resistant to early blight and improve our understanding of the mechanisms that underly early blight resistance.

## 2. Materials and Methods

### 2.1. Plant Materials

*Solanum* genotypes used in this study are listed in Table 1. All plants were maintained in tissue culture on MS20 medium (4.4 g Murashige and Skoog basal salt mixture including vitamins, 20 g sucrose and 8 g/L micro agar, pH = 5.8) [30]. Fresh shoots were propagated two weeks prior to transferring plants to soil. Plants were grown in greenhouses of Unifarm (Wageningen University & Research) under long-day conditions (16 h light/8 h dark).

### 2.2. Generation of Populations

Population AJW5 was generated by pollinating emasculated flowers from RH89-039-16 with pollen from BER 491-1, and population AJW8 was generated by pollinating flowers from MLM 266-2 with pollen from RH89-039-16. Ripe berries were harvested about 6 weeks after pollination. Seeds were harvested from the ripe berries, washed with tap water and dried at room temperature on filter paper for 2 weeks. Seeds were then sterilized by washing them in 70% ethanol, followed by a 15 min incubation in a 1.2% sodium hypochlorite solution and washed once more in 70% ethanol. Finally, the seeds were rinsed 3 times in sterilized tap water, sown on MS20 medium and incubated in the dark until germinated.

### 2.3. Ploidy Determination

A leaf sample from MLM 266-2, RH89-039-16 and each progeny genotype of the cross between these genotypes was sent to Plant Cytometry Services (Didam, The Netherlands) for ploidy determination using flow cytometry and 4′,6-diamidino-2-phenylindole (DAPI) staining.

### 2.4. Production of Inoculum from A. solani

Spores from *A. solani* isolate altNL03003 (CBS 143772) were produced as described in Wolters et al. (2019) [16]. The isolate was maintained on potato dextrose agar (PDA) [31]. Small plugs containing mycelium of *A. solani* were transferred to autoclaved V8 medium (5x diluted V8 juice) and incubated in a shaking incubator at 28 °C for five days. The resulting culture was poured into PDA Petri dishes and incubated without lids in an incubator equipped with blacklight fluorescent tubes (Philips TL-D 18 W BLB) at 25 °C (12 h dark/12 h blacklight) for 3 days. The Petri dishes containing spores were covered and kept in the dark at 25 °C for about a week. Inoculum was prepared by pouring tap water into the Petri dishes and gently scraping conidia from the plates using a plate spreader. The conidial suspension was filtered through a tea strainer and collected in a 50 mL tube. Spore concentration was calculated using a haemocytometer. Conidia were left to settle to the bottom of the 50 mL tube and resuspended in ½ strength potato dextrose broth supplemented with 0.3% micro-agar to obtain a final suspension of 1 × 10^5^ conidia/mL.

### 2.5. Early Blight Testing

Two days before the inoculation, plants were transferred to a climate cell to favour early blight development (25 °C, RH = 70% and 16 h light/8 h dark) [2,16,32]. The plants were placed inside a transparent plastic tent that also contained an atomiser (Condair 505). Five replicates per genotype were included in the test of *Solanum* genotypes and three replicates per genotype were included in the progeny screen. Genotypes were equally divided over three blocks to maintain comparable distances to the atomiser. The first three fully expanded leaves from the top (‘upper’, ‘middle’ and ‘lower’) were each inoculated with six 10 μL droplets of spore suspension. Lights were switched off and the atomiser was turned on after inoculation. The next morning, the normal light regime was resumed, and the atomiser was only switched on during the night. Lesion diameters were measured 5 days post inoculation using a digital calliper. Chlorotic areas around lesions were not included in the measurement. When considering more irregular lesions, we tried our best to estimate the average diameter of a circle matching a similar area. Data were analysed and figures were generated using the ggplot2 package [33] and Rstudio (R Version 4.02) [34,35].

## 3. Results

### 3.1. Genotypes with Promising Levels of Resistance to Early Blight Are Identified in Wild Solanum Germplasm

Previously, we described a rapid method to screen wild *Solanum* for resistance to early blight [16]. In order to select the most promising resistant genotypes for subsequent studies, we used the screening to evaluate a set of *Solanum* genotypes from the germplasm collection of plant breeding at Wageningen University & Research (Table 1). ANOVA showed that there are significant differences between the average lesion sizes of the genotypes that were tested (*p* < 0.05). A Tukey’s HSD test was performed to see which genotypes were different (α = 0.05) and the results are summarized in Figure 1. Several promising resistant genotypes were identified. No signs of infection were observed on the most resistant genotype, *S. commersonii* subsp. *malmeanum* (MLM) 266-2, although we occasionally observed small dark specks reminiscent of a hypersensitive response (Figure 2a,b). Very small, restricted lesions were observed on leaves from *S. berthaultii* (BER) 491-1 (Figure 2c), and *Solanum polyadenium* (PLD) 207-1 and 782-9 showed slightly larger lesions. The *S. tuberosum* dihaploid G254, diploid RH89-039-16 (Figure 2d) and tetraploid cultivars Bintje and Désirée were all susceptible, as well as *S. microdontum* subsp. *gigantophyllum* (GIG) 715-1. We selected MLM 266-2 and BER 491-1 for further studies because both appear much more resistant than any of the *S. tuberosum* genotypes that were tested.

### 3.2. Resistant Wild Solanum Genotypes Can Be Crossed to Susceptible S. tuberosum

To investigate the possibility of transferring resistance of BER 491-1 (2 EBN, Table 1) and MLM 266-2 (1 EBN) to *S. tuberosum*, we attempted crosses with susceptible *S. tuberosum* genotypes G254 and RH89-039-16 (both 2 EBN, Table 2). As could be expected based on compatible EBN, plump berries were obtained without difficulty for the crosses with BER 491-1, which were full of seeds. It proved to be more difficult to obtain seeds from the crosses between MLM 266-2 and *S. tuberosum* genotypes G254 and RH89-039-16. Berries were only obtained in cases where MLM 266-2 was used as a mother and the berries that were found were often empty. Seeds were only obtained in cases where RH89-039-16 was used as a father. On average, about three berries were needed to obtain a single seed.

### 3.3. Progeny from MLM 266-2 ×RH89-039-16 Is Triploid

*S. commersonii* is known to spontaneously produce unreduced female gametes, which can be successfully fertilized by pollen from 2 EBN species [24]. This would explain why few seeds are obtained in the cross between MLM 266-2 and RH89-039-16. To test if the resulting MLM *× S. tuberosum* hybrid seeds were indeed derived from 2n egg cells, we determined their ploidy level using flow cytometry. Seeds were sown in vitro and 24 progeny plants were obtained. We confirmed that MLM 266-2 and RH89-039-16 were diploids and that the progeny (except for a single tetraploid genotype) was triploid (Table 3). The triploid progeny showed a remarkable vigour, as illustrated by the size and dark green colour of the leaves in Figure 3. Thus, we took advantage of the spontaneous formation of 2n egg cells in *S. commersonii*, using it as a mother in a cross with RH89-039-16, to obtain MLM *× S. tuberosum* hybrids.

### 3.4. Resistance to A. solani Is Inherited in Two Distinct Modes in Progeny of Wild Solanum × S. tuberosum

To gain insight into the genetics underlying early blight resistance of BER 491-1 and MLM 266-2, 19 progeny plants from the crosses RH89-039-16 × BER 491-1 (population ‘AJW5’) and 22 progeny genotypes from MLM 266-2 × RH89-039-16 (population ‘AJW8’) were used in a disease test with *A. solani*. Rooted in-vitro-grown plantlets were transferred to the greenhouse and inoculated with spores from *A. solani* as 5-week-old plants. Interestingly, we observed two distinct inheritance patterns of resistance in the two populations.

In the AJW5 population, resistant genotypes, with lesion sizes comparable to resistant parent BER 491-1, as well as susceptible genotypes, with similar lesion sizes as the susceptible parent RH89-039-16, were present among the progeny (Figure 4a). However, most genotypes showed intermediate lesion sizes, with a gradual increase from small to big lesion sizes. The size of the lesions was influenced by the age of the leaf, as lower leaves generally showed bigger lesions than upper leaves. When the distribution of all lesion sizes was shown in a histogram, it became clear that most lesions were of an intermediate size, with comparatively few small or large lesions (Figure 4b). These results point to a quantitative inheritance of resistance to *A. solani* in population AJW5 that was derived from resistant parent BER 491-1.

The resistance pattern in population AJW8 contrasts with that of the AJW5 population. While genotypes with small (<4 mm) lesions as well as large (>6 mm) lesions were found among the progeny from resistant MLM 266-2, no genotypes with mostly intermediate lesion diameters (between 4 and 6 mm) were identified in this population (Figure 4c). When visualizing lesion sizes in a histogram, a bimodal distribution of lesion sizes emerges, with mainly small or large sizes and comparatively few intermediate sizes (Figure 4d). We classified the 12 genotypes with average lesion sizes smaller than 4 mm as resistant and the remaining 10 genotypes with average lesions diameters larger than 6 mm as susceptible. The observed segregation in population AJW8 does not significantly differ from a 1:1 ratio (X^2^ (1, N = 22) = 0.182, *p* = 0.67), suggesting that resistance is controlled by a single major locus in MLM 266-2.

## 4. Discussion

We identified promising resistance against early blight in *S. commersonii* and *S. berthaultii*. Resistant genotypes from both species were successfully crossed with diploid *S. tuberosum*. Interestingly, two patterns of inheritance were observed in progeny from these crosses. Resistance appeared to inherit quantitatively in the progeny from resistant parent BER 491-1. In contrast, resistance in the progeny from MLM 266-2 segregated in a qualitative manner. We found an approximate 1:1 segregation ratio, which suggested that a single dominant gene was responsible for resistance in MLM 266-2.

Traditionally, resistance against necrotrophic pathogens such as *A. solani* is considered to be a complex, polygenic trait, contrasting with the gene-for-gene resistance that is typically described for biotrophic pathogens [38,39,40,41]. Resistance to early blight in potato is indeed usually described as quantitative [6,27,28,29], but detailed studies on the genetics have not been performed in wild species so far.

In recent years, it has become apparent that resistance against necrotrophs is not necessarily always quantitatively inherited. According to the inverse gene-for-gene model, immune receptors can function as susceptibility genes for necrotrophic pathogens, where recognition of an effector leads to susceptibility, and where absence of the effector or host target will result in resistance [42,43,44,45,46]. Susceptibility is inherited dominantly in this case, while resistance is recessive, which seems to contrast with the dominant resistance identified in MLM 266-2. It is not the first time that such resistance has been observed though. A major dominant gene was found to contribute to resistance against necrotrophs, including *Alternaria brassicicola* and *Alternaria brassicae,* in Arabidopsis [47]. More recently, it was found that a nucleotide-binding/leucine-rich repeat (*NLR*) gene can provide resistance to *A. alternata* in apple, based on the recognition of an effector [48]. This shows that classical *NLR* gene-mediated resistance can be effective against *Alternaria*.

*S. commersonii* and *S. berthaultii* (or *S. tarijense* [49]) have both previously been described as some of the most early blight-resistant *Solanum* species [17,50]. Resistance observed in MLM 266-2 looked especially promising, since we saw no early blight symptoms on this genotype, apart from some occasional dark specks that seem reminiscent of a hypersensitive response. Slightly larger lesions were observed in the offspring from the MLM 266-2 × RH89-039-16 cross, but it is important to notice that lesion diameters did not exceed 4 mm, which was about the diameter of the inoculation droplet. The difference between the most resistant progeny and resistant parent MLM 266-2 could be explained by the fact that factors from *S. tuberosum* attenuated the response from MLM 266-2. Still, the approximately 1:1 segregation of resistance indicated a major contribution from a single locus in MLM 266-2.

We demonstrated here that resistance against early blight from wild *Solanum* can be transferred to diploid *S. tuberosum* through conventional crosses. The most resistant progeny genotypes described in this study provide a starting point for the development of a potato cultivar that is naturally resistant to early blight. Resistance against late blight caused by *P. infestans* has previously been introgressed from wild *Solanum* in *S. tuberosum* germplasm and various *R* genes against this particular disease have already been identified in wild relatives of potato, including *S. berthaultii* [51]. The availability of molecular markers linked to resistance greatly facilitates this process. The cross between BER 491-1 and *S. tuberosum* yielded many seeds and the population forms a good basis for a study to find quantitative trait loci (QTL) and to develop markers linked to early blight resistance. So far, such studies have only been performed in cultivated potato and not in wild *Solanum* [28,52].

The observed segregation of resistance in the *S. commersonii*-derived population AJW8 indicates that the locus that causes resistance is heterozygous in MLM 266-2. Because of different EBN classes, progeny are only obtained when MLM 266-2 is used as a mother in crosses with *S. tuberosum*, as a result of the spontaneous formation of unreduced egg cells [24]. In potato, 2n egg cell formation is mostly caused by second-division restitution. The resulting gametes are largely homozygous because the sister chromatids are not separated. However, heterozygosity increases towards the telomeres, because of recombination [53,54]. If resistance is caused by a single dominant gene in MLM 266-2, then both homozygous resistant as well as heterozygous gametes from MLM 266-2 would yield resistant offspring. *S. commersonii* has several interesting properties besides early blight resistance, such as frost tolerance and resistance to other diseases [55]. Several approaches have been proposed to produce hybrids between *S. commersonii* and *S. tuberosum* despite their different EBNs [24,56,57] and identification of the gene that causes resistance would help the development of early-blight-resistant potato cultivars through conventional breeding as well as gene technology.

In a follow-up study, it would be good to test additional *Alternaria* isolates on the resistant genotypes that were identified in this study, to make sure that the resistance is of practical use. In addition, it would be interesting to see if genotypes that are susceptible to *A. solani* are present among *S. commersonii* or other compatible 1 EBN *Solanum* and if a similar segregation of resistance is found in progeny derived from crosses with these genotypes. It would be more practical to continue a mapping study in such a population, especially since many genotypes are required when aiming to fine-map or identify the locus or gene underlying resistance in MLM 266-2. A genome sequence is already available for *S. commersonii* [55], which can aid the development of molecular markers and the identification of candidate resistance genes. However, the genome is rather fragmented and it is not clear if the genotype that was sequenced is resistant to *A. solani*. Approaches such as BSA-RNAseq, *R* gene enrichment sequencing or derivative enrichment sequencing technologies such as receptor-like protein (RLP) and receptor-like kinase (RLK) sequencing [58,59] could provide a short-cut to identifying the causal gene in case resistance of MLM 266-2 is based on an immune receptor.

## 5. Conclusions

We provide evidence for quantitative as well as qualitative resistance to early blight disease in wild *Solanum*. Moreover, we show that resistant hybrids can be obtained in crosses of *S. commersonii* and *S. berthaultii* with *S. tuberosum*. This research is a step towards a better understanding of the mechanisms that can provide resistance to *A. solani*. It can contribute to the development of a potato cultivar with natural resistance, paving the way for a more sustainable control of early blight disease in potato.

## Figures and Tables

**Figure 1 biology-10-00892-f001:**
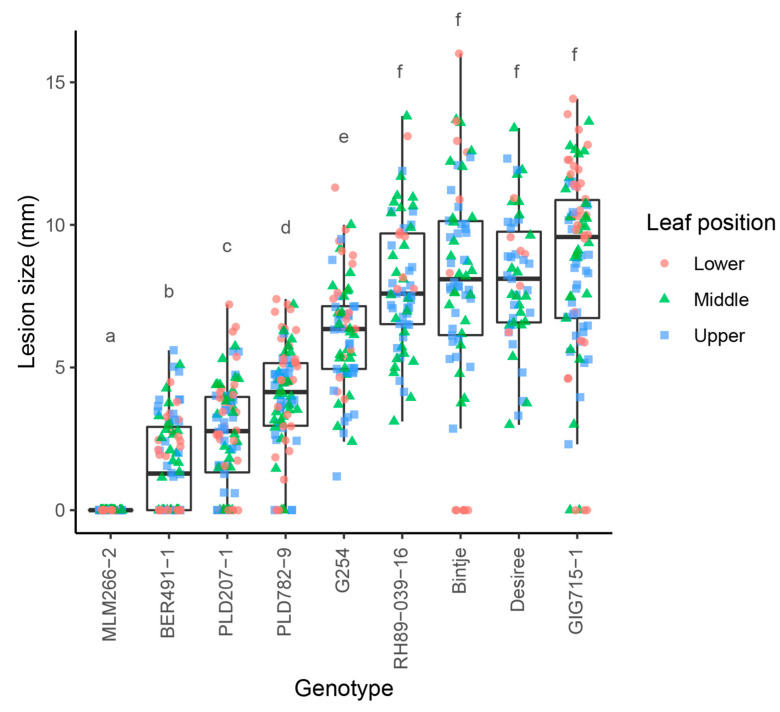
Genotypes with promising levels of resistance to early blight are identified in wild *Solanum* germplasm. *Solanum* genotypes (Table 1) were inoculated with *Alternaria solani* isolate altNL03003 and lesion diameters were measured 5 days post inoculation. Lesion sizes are visualised with boxplots, with horizonal lines indicating median values and individual measurements of lower (red), middle (green) and upper (blue) leaves plotted separately in the figure. Different letters are used to indicate significant differences between the lesions of different genotypes (Tukey’s HSD test, α = 0.05).

**Figure 2 biology-10-00892-f002:**
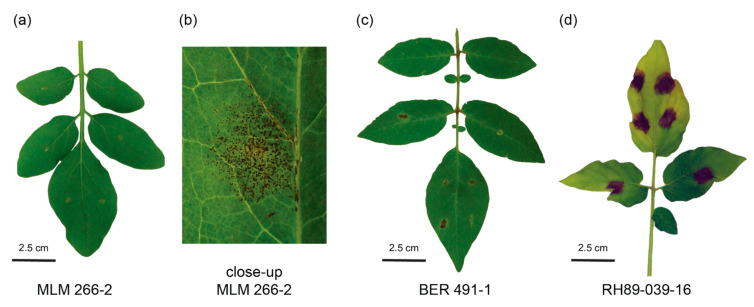
Disease symptoms on different *Solanum* genotypes. Pictures were taken from representative leaves 5 days post inoculation of *S. commersonii* subsp. *malmeanum* (MLM) 226-2 (**a**) and in close-up (**b**), *S. berthaultii* (BER) 491-1 (**c**) and *S. tuberosum* RH89-039-16 (**d**).

**Figure 3 biology-10-00892-f003:**
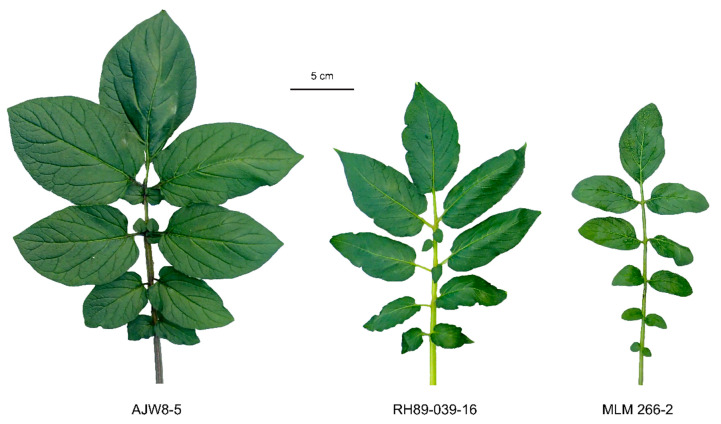
Enhanced vigour of triploid progeny genotype AJW8-5 from diploid *S. commersonii* subsp. *malmeanum* (MLM) 266-2 ×diploid *S. tuberosum* RH89-039-16. Pictures are taken from the first fully expanded leaf from the top of 8-week-old plants. A leaf from representative progeny genotype AJW8-5 is shown next to leaves from parents RH89-039-16 and MLM 266-2.

**Figure 4 biology-10-00892-f004:**
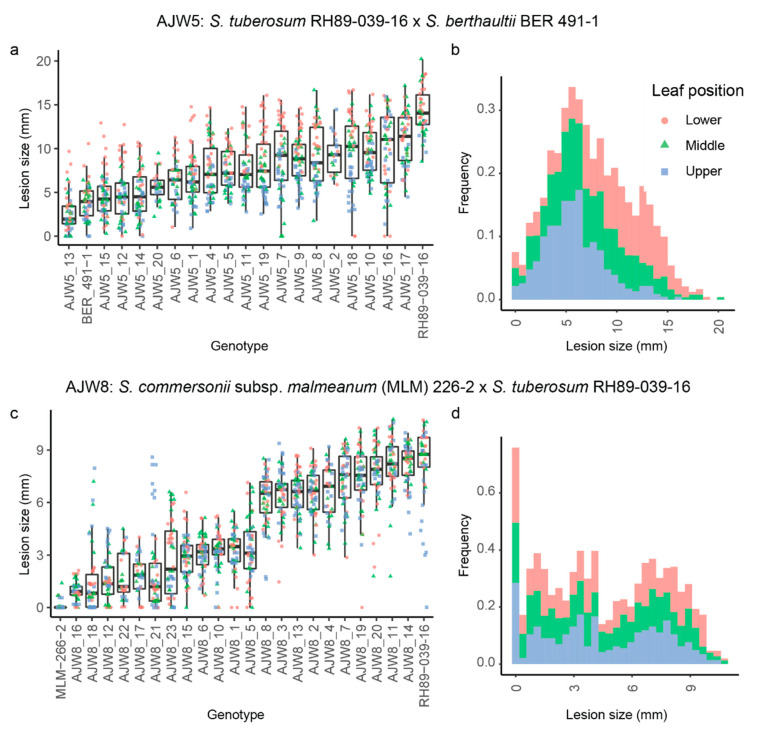
Quantitative and qualitative inheritance of early blight resistance in progeny of wild *Solanum*. Inheritance of early blight resistance was assessed by inoculating parents and progeny from the AJW5 population (*S. tuberosum* RH89-039-16 × *S. berthaultii* BER 491-1) and the AJW8 population (*S. commersonii* subsp. *malmeanum* 226-2 × RH89-039-16.) Lesions measured on lower (red), middle (green) and upper (blue) leaves are visualised using boxplots (**a**,**c**) and the distribution of lesion sizes is shown in histograms (**b**,**d**).

**Table 1 biology-10-00892-t001:** Overview of genotypes used in this study. Species, genotype name, ploidy and EBN class of each genotype are listed.

Scheme	Genotype	Ploidy	EBN
*Solanum commersonii* subsp. *malmeanum* (MLM)	266-2	2×	1 EBN
*Solanum berthaultii* (BER)	491-1	2×	2 EBN
*Solanum polyadenium* (PLD)	207-1	2×	2 EBN
*Solanum polyadenium* (PLD)	782-9	2×	2 EBN
*Solanum tuberosum*	G254 ^a^	2×	2 EBN
*Solanum tuberosum*	RH89-039-16 ^b^	2×	2 EBN
*Solanum tuberosum*	Bintje	4×	4 EBN
*Solanum tuberosum*	Désirée	4×	4 EBN
*Solanum microdontum* subsp. *gigantophyllum* (GIG)	715-1	2×	2 EBN

^a^ Olsder et al. (1976) [36]; ^b^ Xu et al. (2011) [37].

**Table 2 biology-10-00892-t002:** Resistant wild *Solanum* genotypes can be crossed to susceptible *S. tuberosum*. Selected resistant wild *Solanum* genotypes *S. commersonii* subsp. *malmeanum* (MLM) 226-2 (1 EBN) and *S. berthaultii* BER 491-1 (2 EBN) were crossed with susceptible *S. tuberosum* genotypes RH89-039-16 and G254 (2 EBN). The number of berries and seeds obtained for each cross are listed.

Cross	Mother	×	Father	Berries	Seeds
AJW5	RH89-039-16 (2 EBN)	×	BER 491-1 (2 EBN)	11	~400
AJW6	G254 (2 EBN)	×	BER 491-1 (2 EBN)	18	~1000
AJW8	MLM 266-2 (1 EBN)	×	RH89-039-16 (2 EBN)	166	59
AJW9	MLM 266-2 (1 EBN)	×	G254 (2 EBN)	20	0

**Table 3 biology-10-00892-t003:** A cross between diploid *Solanum commersonii* subsp. *malmeanum* (MLM) 266-2 and *S. tuberosum* RH89-039-16 (population ‘AJW8’) yields triploid progeny. Ploidy levels of 20 progeny genotypes were established through flow cytometry, using both diploid parents as a reference. All progeny were found to be triploid (3×), except genotype AJW8-11 (4×).

Genotype	Ploidy
AJW8-1	3×
AJW8-2	3×
AJW8-3	3×
AJW8-4	3×
AJW8-5	3×
AJW8-6	3×
AJW8-7	3×
AJW8-8	3×
AJW8-10	3×
AJW8-11	4×
AJW8-12	3×
AJW8-13	3×
AJW8-14	3×
AJW8-15	3×
AJW8-16	3×
AJW8-17	3×
AJW8-18	3×
AJW8-20	3×
AJW8-21	3×
AJW8-22	3×
MLM 266-2	2×
RH89-039-16	2×

## Data Availability

Datasets supporting the findings of this study are available from the corresponding author upon reasonable request.

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
