# Peer review of "Qualitative and Quantitative Resistance against Early Blight Introgressed in Potato"

_biology, 2021, doi:10.3390/biology10090892_

Round 1

Reviewer 1 Report

The manuscript entitled: `Qualitative and quantitative resistance against early blight in trogressed in potato´ deals with a very interesting and topical subject. The search of cultivars resistant to fungal diseases such as early blight is a sustainable solution, and in many regions of the world it is probably the alternative to the abuse of fungicides for its control. In recent years, the increase in global temperature, and the increase in stormy events favor the environmental conditions for the increase of diseases and pests of agricultural crops.

In general, the presented manuscript is interesting, the experiments were well planned and performed. The results are consistent with the stated objective, and the discussion is acceptable. Although I recommend statistical treatment of the data to provide quality and scientific rigor to the manuscript.

Therefore, I recommend considering the manuscript for publication with few considerations:

- In title, I don't know what the authors mean by the word `trogressed´ of the title.

- The number of samples (plants) used for the study to be included. You can include it in Table 1.

- In line 104- 105, the conditions must be referenced. There are several researchers who reported the optimal temperature and humidity conditions for the development of early blight.

- The text of legend of Figure 1: `The first three fully expanded leaves from the top (‘upper’, ‘middle’ and ‘lower’) were each inoculated with six 10 μl droplets of spore suspension and three replicates we used for each genotype. Lesion diameters were measured 5 days post inoculation ´, I think it is better to include it in the materials and methods section. Actually in the figure the results are shown.

- The information included in the legend of Figure 2 refers to the results of the symptoms, it is better to incorporate it into the text itself. The legend should reflect a short explanation, not results. Same for Figure 3.

- A statistical treatment could be applied with the data of the lesions (in mm), for example an ANOVA, showing the differences and similarities of the genotypes (P < 0.05).

Reviewer 2 Report

The article concerns identification of resistance to Alternaria in wild Solanum relatives of potato.  The study is well done and include characterization of unreduced gametes from S. commersonii.

I have one additional comment. Authors should add information to the Materials and Methods as follows:
Line 113 Indicate version of R used and the citation for the R software.

Reviewer 3 Report

The manuscript "Qualitative and quantitative resistance against early blight introgressed in potato" reported two successful introgression of wild potato, Solanum berthaultii (BER) and Solanum commersonii subsp. malmeanum (MLM) with domesticated potato Solanum tuberosum (RH89-039-16), respectively. The former population AJW5 exhibited a quantitative inheritance in early blight resistance, and surprisingly, the later population AJW8 exhibited a qualitative resistance. The rationale is well expressed and the text is well written. There are some parts that I would suggest the authors to improve:

(1) The authors used one fungal isolate in this study. However, is it possible that other isolates may infect the BER or MLM, and is it possible wild potatoes may have theirs native Alternaria pathogens to cause early blight? A simple experiment is to inoculate the BER, MLM, and RH89-039-16 with different isolates to ensure the  early blight resistance can be observed in different isolates. The authors may also  provide more background regarding the diseases of BER and MLM, and whether the introgression may introduce better resistance to isolates infecting domesticated potato, but bring potential susceptibility to isolates infecting BER or MLM?

(2) The authors measured the lesion diameters (line 111) to represent the disease severity. However, the details regarding how many lesions were measured, how the authors handle the irregular shapes of lesions, and if any chlorosis halo were observed or included in the diameter was not clearly described. It is recommended that the authors may provide more descriptions in the measurements.

(3) The authors described the result of AJW8 to be dominantly inherited, which was surprisingly different from the conventional inverse gene-for-gene model to necrotrophic pathogens that the resistance is recessive. However, the inverse gene-for-gene model depicts the toxin-inducing symptoms, and the study inoculated the plants with fungal spores. As the both populations were relatively small (24 progenies) and the lesion measurement in size may not provide any information regarding the susceptibility to phytotoxicity or necrotrophic effectors, the authors may want to be conserved. In addition to reverse gene-for-gene model, another possibility could be loss-of-susceptibility that is generally qualitative inherited. It would be better to include the possibility in the discussion. 

line 42, please spell out EBN

line 227, do you mean ratio instead of ration?
